# Fabrication of Reusable Carboxymethyl Cellulose/Graphene Oxide Composite Aerogel with Large Surface Area for Adsorption of Methylene Blue

**DOI:** 10.3390/nano11061609

**Published:** 2021-06-18

**Authors:** Wei Zhu, Xueliang Jiang, Kun Jiang, Fangjun Liu, Feng You, Chu Yao

**Affiliations:** 1School of Materials Science and Engineering, Wuhan Institute of Technology, Wuhan 430205, China; witzhuwei@163.com (W.Z.); jk95922@126.com (K.J.); 15387182689@wit.edu.cn (F.L.); youfeng.mse@wit.edu.cn (F.Y.); chuyao@wit.edu.cn (C.Y.); 2School of Chemical and Materials Engineering, The College of Post and Telecommunication of WIT, Wuhan 430073, China; 3Hubei Key Laboratory of Plasma Chemistry and Advanced Materials, Wuhan Institute of Technology, Wuhan 430205, China; 4Key Laboratory of Green Preparation and Application for Functional Materials, Ministry of Education, Hubei University, Wuhan 430062, China

**Keywords:** adsorption, aerogel, carboxymethyl cellulose, graphene, methylene blue, reusability

## Abstract

A highly efficient adsorbent for methylene blue (MB) adsorption was developed by combination of carboxymethyl cellulose (CMC) and graphene oxide (GO) via a simple one-step hydrothermal method. The as-synthesized CMC/GO composite aerogel has a mesoporous structure with an average pore diameter of 30 nm and a high specific surface area of 800.85 m^2^·g^−1^. Moreover, the CMC/GO composite aerogel demonstrates a significant selectivity for the dye adsorption, especially for MB, where its adsorption capacity can reach 244.99 mg·g^−1^ with an excellent recyclability for more than nine times. Thus, the prepared CMC/GO composite aerogel would be an effective adsorbent for dyes adsorption, owing to the merits of high efficiency, reusability, and eco-friendliness.

## 1. Introduction

The industrial effluents caused serious environmental problems, especially for effluents containing anionic [1,2] and cationic [3,4,5] dyes. In the past few years, many researches have been carried out to solve this industrial effluents pollution issues. The proposed methods to treat dye-contaminated wastewater mainly include reverse osmosis, coagulation, chemical precipitation, membrane separation, solvent extraction, adsorption, and catalytic degradation [6,7,8,9,10]. Among the aforementioned techniques, adsorption is proved to be the most effective and promising technique to overcome this serious environmental problem [11].

As adsorbents, materials with high specific surface area play an important role in the adsorption of pollutants. Graphene aerogels are widely used in water pollution treatment due to their high porosity (99.8%) and high specific surface area [12]. Graphene can react with a variety of functional materials to prepare composites, increasing the type and number of functional groups, which can be used to adsorb pollutants. Many related studies have been reported.

For instance, Zhang et al. [13] fabricated a novel three-dimensional (3D) graphene hydrogel using L-cysteine as both templating and reducing agent. The specific surface area of the hydrogel was 154 m^2^·g^−1^, and the adsorption capacity for methyl orange (MO) was 75 mg·g^−1^. Zhao et al. [14] synthetized carboxylated graphene oxide/chitosan/cellulose composite beads (GCCSC) by a sol-gel method for the adsorption of Cu^2+^ (22.4 mg·g^−1^). Zhang et al. [15] synthetized a three-dimensional reduced graphene oxide and montmorillonite composite aerogel with specific surface area (43.17 m^2^·g^−1^), by a sol-gel method for the adsorption of Cr^6+^ (77.52 mg·g^−1^). Zhao et al. [16] prepared graphene oxide/Poly (ethyleneimine) 3D aerogel by a chemical crosslinking method, with high specific surface area (526.4 m^2^·g^−1^) and high adsorption capacity for MB (249.6 mg·g^−1^).

Carboxymethyl cellulose (CMC) is an important active substance that is combined with graphene in the preparation of composites. It is a water-soluble cellulose derivative containing carboxyl groups, and widely used as a suitable agent for wastewater treatment. By combining GO with CMC, a three-dimensional composite aerogel can be prepared to achieve complementary advantages and synergistic effects [17]. Zhang et al. [18] constructed multiaperture graphene oxide/carboxymethyl cellulose (GO/CMC) monoliths adopting a unidirectional freeze-drying technique, with high adsorption ability for Pb^2+^ (76.7 mg·g^−1^). Chen et al. [19] synthesized magnetic cellulose-supported GO composite (Fe_3_O_4_/CMC/GO) by a blending method for the adsorption of Cu^2+^ (199.98 mg·g^−1^).

However, the above methods involve many steps and procedures, and the experimental methods are relatively complex. In our previous research, we have mastered the hydrothermal synthesis approach and produced a series of new materials. For instance, zinc-doped nickel oxide hollow microspheres (Zn-NiO-HM) with good electrochemical properties [20], mono-dispersed ceria (CeO_2_) hollow nanospheres for removing Cr^6+^ [21], and ytterbium oxide (Yb_2_O_3_) hollow microspheres with core-shell structure [22] have been successfully synthesized. Furthermore, the low cost, high yield, easy operation, scalability, and controllability of the hydrothermal method make this synthesis approach suitable for the large-scale production of graphene aerogels.

In this study, carboxymethyl cellulose and graphene oxide are used as the main synthetic materials to prepare the CMC/GO composite aerogel for the adsorption of MB through the one-step hydrothermal method. To better understand the characteristics of this aerogel composite, the investigations on the adsorption mechanism, and the establishment of an adsorption model are carried out based on a series of characterizations in structure and properties. The synthesized CMC/GO composite aerogel has high surface area and porous three-dimensional network structure. It has high adsorption capacity and excellent cycle stability for MB, and shows high selectivity for different dyes, which can be widely used in the treatment of wastewater containing dyes.

## 2. Experimental

### 2.1. Chemicals

Carboxymethyl cellulose (CMC) and ethylenediamine (EDA) were purchased from Aladdin Industrial Corporation (Shanghai, China). Natural graphite powder (97%), phosphoric acid (88%), potassium permanganate (99.9%), hydrochloric acid (38%), concentrated sulfuric acid (98%), sodium hydroxide, methylene blue (MB), methyl orange (MO), congo red (CR), and rhodamine B (RhB) were purchased from Sinopharm Chemical Regent Co. Ltd. (Shanghai, China).

### 2.2. Synthesis of Graphene Oxide (GO) Aerogel

GO (Graphene oxide) was prepared by improved Hummers method (with citation about this method) and dispersed in deionized water for 1 h. The 4 mg·mL^−1^ GO dispersion solution was blended with the same amount of EDA, and the blend was then subjected to hydrothermal reaction at 120 °C for 12 h. After this, graphene oxide (GO) hydrogel was self-assembled by hydrothermal reaction. The hydrogels were rinsed by deionized water for 1–2 times, and then freeze-dried at −50 °C for 48 h. Finally, three-dimensional GO aerogel was obtained.

### 2.3. Synthesis of Carboxymethyl Cellulose/Graphene Oxide (CMC/GO) Composite Aerogel

The GO dispersion solution (4 mg·mL^−1^) was added to a certain amount of EDA and stirred for 1 h until completely dissolved. CMC with different qualities were added into the above solution so that the mass ratio of GO: CMC was 2:1, 1:1, and 1:2. After that, the mixed solution was transferred into a hydrothermal reactor to react with 12 h at 120 °C to produce the CMC/GO composite hydrogel. Finally, the hydrogel was placed in a freeze dryer (−50 °C) for 48 h, and the CMC/GO composite aerogel was produced.

### 2.4. Material Characterizations

In this study, Fourier transform infrared (FTIR) spectroscopy (Thermo Electron Nicolet 6700 spectrometer, Thermo Electron Corporation, Waltham, MA, USA) was used to test the functional groups of RGO and CMC/RGO aerogels. Powder X-ray diffraction (XRD, Bruker D8-Advance, 20 mA, Bruker, Karlsruhe, German) was used to determine the phase composition. The morphology and structure of the aerogels were studied by scanning electron microscope (SEM, JSM-5510LV, JEOL, Tokyo, Japan) and transmission electron microscope (TEM, JEM-2100, JEOL, Tokyo, Japan). The hydrophobicity of aerogels was measured by the contact angle tester of German Kruss Company (Hamburg, German) DSA100. The aerogel level was put on the sample table and the dripping device was slowly rotated about 10 L droplet to the surface of the sample, then took pictures with the camera on the instrument, and cut the gas liquid interface on the photo. The tangent line was also the contact angle of the water droplet contact surface. The ultraviolet visible-light diffused reflection spectrums (UV-DRS) of the samples were recorded by ultraviolet-visible spectrometer (PerkinElmer, lambda35, Waltham, MA, USA). Nitrogen (N_2_) adsorption-desorption isotherms were obtained at liquid N_2_ temperature using a full automatic specific surface area and aperture distribution analyzer (Autosorb IQ, Quantachrome, Boynton Beach, FL, USA).

### 2.5. Adsorption Characterization

Through determining the adsorption rate of MB solution, the adsorption activity of the sample was evaluated. A certain amount of CMC/GO composite aerogel was added in 50 mL MB dye solution and the pH was adjusted from 1 to 11. After the fixed time, the aerogel and solution were separated at once. The concentration of MB solution was determined by UV-DRS. The absorption rate (*R*%) was calculated by Equation (1) and the adsorption capacity (*q_t_*) was calculated by Equation (2).
(1)$$R=\frac{C_{0}-C_{t}}{C_{0}}\times 100\%$$
(2)$$q_{t}=\frac{C_{0}-C_{t}}{m}V$$

## 3. Results and Discussion

### 3.1. Characterization Results

The chemical features of the functional groups of the prepared samples are investigated using FTIR, and the results are shown in Figure 1A. For GO aerogel, the peaks at 3432, 1728, 1624, and 1060 cm^−1^ are attributed to the stretch vibration of O-H, C=O, C=C (aromatic ring), and C=O (epoxy) bonds [23,24], respectively. After combining with CMC, the CMC/GO composite aerogel sample shows the absorption bands at 3455, 2920, 1580, 1402, and 1048 cm^−1^, which are related to the stretch vibrations of O-H, C-H, asymmetric and symmetric stretching vibrations of C=O, and the bend vibration of C-O-C [25], respectively. The comparison of FTIR spectra shows that CMC/GO composite aerogel contains abundant hydrophilic groups such as hydroxyl group, carboxyl group, and epoxy group, which have good chemical activity and are beneficial to the adsorption of cationic dyes.

Figure 1B shows the XRD spectra of CMC, GO aerogel, and CMC/GO composite aerogel. It is obvious that the CMC aerogel shows typical diffraction peaks at 19.93°, which is attributed to CMC crystal shape. The diffraction peak of the GO aerogel appears at 25.05°, while the CMC/GO composite aerogel has a weak diffraction peak at 23.58°. The interlayer spacing in GO aerogel calculated by Bragg equation is 0.36 nm, while the interval in CMC/GO composite aerogel is 0.38 nm, which shows a slight increase. Compared with the GO aerogel, the diffraction peak of CMC/GO composite aerogel shifts to the left and the peak strength is weaker. Nevertheless, the structural stacking of CMC/GO composite aerogel is more porous. Due to the introduction of CMC, the interlayer spacing of aerogel increases somewhat, indicating that the graphene lamellae may be rearranged.

Figure 1C,D display the N_2_ adsorption and desorption curves of GO aerogel and CMC/GO composite aerogel. What is more, the Table 1 reports the Brunauer–Emmett–Teller (BET) specific surface of the CMC/GO composite aerogel reaches 800.85 m^2^·g^−^^1^, which is obviously greater than the 65.01 m^2^·g^−^^1^ obtained by GO aerogel. When CMC is grafted on the surface of graphene, the distance between graphene layers is increased and denser mesoporous is formed. Meanwhile, the specific surface area increases. Similarly, the adsorption point on the surface of aerogel is increased and the contact probability between dye molecules and adsorption sites is increased, which improves the adsorption performance of CMC/GO composite aerogel.

Figure 2A is the SEM image of the large and dispersed aerogel lamellae of the GO aerogel prepared by EDA as reductant. Figure 2B shows SEM image of CMC/GO composite aerogel. CMC/GO composite aerogel is distributed in slices with thin edges, obvious fluctuations, and tubular pore structures formed by the cross-linking of the slices. The lamellae are not simply overlapped and stacked in an orderly way, but cross linked and supported each other through overlapped, crossed and other forms, forming a porous three-dimensional network structure. The pore size of CMC/GO composite aerogel decreases and the specific surface area increases after the combination of CMC, which makes the adsorption sites more accessible to dye molecules, and is conducive to the adsorption of MB. Figure 2C,D shows TEM image of CMC/GO composite aerogel. As shown in the TEM images, CMC/GO composite aerogel has porous structure with uniform pore distribution and good morphology. This proves that the hydrothermal method successfully prepared CMC/GO composite aerogel with three-dimensional porous structure, which can significantly increase the specific surface area and thus improve the adsorption performance.

As shown in Figure 3A,B, the advancing contact angle of GO is 65.8°, and the receding contact angle is 45°. By calculation, the hysteresis value is 20°, indicating that GO aerogel surface is not uniform, which is consistent with the SEM image. Additionally, the contact angle of GO aerogel is acute and less than 90^°^, showing moderate hydrophilicity. In Figure 3C, the contact angle of CMC/GO composite aerogel is almost 0°, indicating strong hydrophilicity. This is due to the large number of carboxyl groups in CMC. The swelling phenomenon of CMC/GO aerogel is observed obviously after water absorption.

### 3.2. Adsorption Performance of MB on CMC/GO and GO Aerogels

At room temperature, 50 mL of 50 mg·L^−1^ MB solutions stirring for 3 h with 10 mg of RGO aerogel and CMC/GO composite aerogel, respectively. Figure 4A shows that the adsorption efficiency of CMC/GO composite aerogel (97.99%) is obviously larger than the aerogel obtained by GO (48.07%). Indeed, GO aerogel contains generous amino groups on the surface, which makes it poor adsorbent of cationic dyes due to their electrostatic interaction, so its adsorption rate for MB merely attains 48.07%. However, the introduction of CMC, endows multiple holes on the GO surface as well as enormous number of functional groups such as carboxyl, hydroxyl, and epoxy groups. That consequently accelerate the spreading of dyes into the adsorbents and increased adsorption property of CMC/GO composite aerogel, and the adsorption capacity is 244.99 mg·g^−1^.

### 3.3. Effect of Adsorption Performance

Different mass ratio is tried (2:1, 1:1, and 1:2) to study the influence of CMC and GO proportion in the MB adsorption. As shown in Figure 4B, when the mass ratio of CMC/GO is 2:1, the adsorption effect of MB is the best, with an adsorption efficiency of 97.2% at 180 min. This result indicates the synthesized aerogel with the CMC/GO mass ratio of 2:1 has the optimal pore microstructure and adsorption sites to accommodate more MB molecules comparing with the samples prepared at other mass ratio.

Furthermore, the effect of pH value on the adsorption performance was studied in the range of 1~11. As can be seen from Figure 4C, the increase of pH value, induces the increment of the MB adsorption rate on the aerogel. This can be explained by the fact that adsorption sites on aerogel are occupied by −H^+^ ions in the acidic solution, resulting in the protonation of −OH and −COOH and the surface charge of aerogels becomes the positively charged. This will create electrostatic repulsion toward the cationic MB and consequently reduces the adsorption capacity. However, when the pH attains the basic region, the deprotonation of -OH and -COOH will occur. The surface of the aerogel will be negatively charged, such as -COOH^−^, and create the appropriate electrostatic attraction to bind the cationic dye. This is conducive to increase the adsorption capacity of MB [26].

The adsorption performance of the prepared aerogels was evaluated by choosing anionic dyes (CR and MO) and cationic dyes (RhB and MB) as target pollutants. Figure 4D shows clearly that the adsorption properties of the aerogel on various dyes are as below: MB > RhB > CR > MO, exhibiting adsorption selectivity to cationic and anionic dyes. The powerful electrostatic attraction (anionic-cationic interaction) among the -COO^−^ groups existed in the CMC/GO composite aerogel and the cationic dye molecule is the major motive force for the adsorption procedure [18]. Towards the anionic dyes (CR and MO), the main driving force is hydrogen bond, leading to the lower adsorption capacity [27]. The difference of adsorption capacity is also related to the strength of the conjugated π-π bond between the aerogel and the groups on the dye molecules, accompanied with the type of charge carried by the dye molecules in the aqueous solution.

### 3.4. Desorption and Cycling Tests

As a promising adsorbent, it requires high adsorption ability for pollutants, and fine renovation property. In general, the used renovation techniques of adsorbent included the oxidation method [28], electrochemical method [29], and solvent method [30]. According to the previous experiments, it was not favorable for CMC/GO aerogel to adsorb MB under acidic conditions. Hence, the solvent method was employed in this work, HCl (10%, *v*/*v*) and ethanol solution (50%, *v*/*v*) were adopted for desorption.

When the adsorption experiments were finished, the aerogels were separated from the remaining MB solution. After washing the aerogel three times with deionized water, the adsorbent was first placed in 100 mL HCl solution and shaken for 1 h. Then the above steps were repeated three to five times with ethanol solution until the solution was almost transparent. The obtained aerogel was cleaned with deionized water to eliminate the residual ethanol solution. Finally, it was dried to constant weight in the oven (60 °C). The dye can be well desorbed from the adsorbent. This is because the carboxyl group on the adsorbent adsorbs MB by electrostatic attraction. Under strong acid condition, high -H^+^ concentration will replace methylene blue molecule, which can desorb MB. Figure 5 exhibits that the adsorption capacity of CMC/GO composite aerogel decreases from 244.99 mg·g^−1^ to 227.02 mg·g^−1^ after nine times of reuse, and the adsorption efficiency decreases from 97.99% to 90.81%, which remains 92.66% at the first time. Compared with the related performance in Table 2, the CMC/GO composite aerogel prepared in this paper has obvious advantages in repeated use, which can be used as an adsorbent with excellent recycling utilization.

## 4. Mechanism Analysis of CMC/GO Composite Aerogel

### 4.1. Isothermal Adsorption Model

The adsorption behavior of CMC/GO composite aerogel on MB was analyzed by Langmuir [36] (Equation (3)) and Freundlich [37] (Equation (4)) adsorption isotherm equation, as shown in Figure 6A,B:(3)$$\frac{C_{e}}{q_{e}}=\frac{C_{e}}{q_{max}}+\frac{1}{k_{L}q_{max}}$$
(4)$$\ln q_{e}=\ln k_{F}+\frac{1}{n}\ln C_{e}$$

From the data, the correlation coefficient (R^2^ = 0.99843) of the Freundlich equation is slightly larger than the Langmuir equation (R^2^ = 0.97678). Therefore, the adsorption of MB on CMC/GO composite aerogel is more consistent with the Freundlich adsorption process and belongs to the heterogeneous surface adsorption process. In addition, the *1/n* value (0.16925) of Freundlich isotherm parameter indicates that CMC/GO composite aerogel is a good adsorbent for MB.

### 4.2. Adsorption Kinetics

Quasi-first-order adsorption kinetics model (Equation (5)) and quasi-second-order adsorption kinetics model (Equation (6)) [38] were used to fit the adsorption data.

Quasi-first-order adsorption kinetic model:(5)$$\ln(q_{e}-q_{t})=\ln q_{e}-k_{1}t$$

Quasi-secondary adsorption kinetic model:(6)$$\frac{t}{q_{t}}=\frac{1}{k_{2}q_{e}{}^{2}}+\frac{t}{q_{e}}$$

Figure 6C,D show the correlation coefficients R^2^ fitted by the quasi first-order adsorption kinetics and the quasi-secondary adsorption kinetics model are 0.99481 and 0.99904, respectively. The calculated maximum adsorption according to the equation is 329.9628 mg·g^−1^ and 264.550 mg·g^−1^. The fitted correlation coefficient of quasi-secondary adsorption kinetic model is relatively higher, and the maximum adsorption capacity calculated by fitting is also closer to the actual value of 244.269 mg·g^−1^, indicating that quasi-secondary adsorption kinetic model can well represent the chemical adsorption [18], which is beneficial for the adsorption process in MB solution.

### 4.3. Adsorption Thermodynamics

Adsorption thermodynamics shows the effect of temperature on the adsorption behavior. The change of system temperature also changes the structural characteristics of the composite aerogel and the movement rate of dye molecules. The adsorption performance of CMC/GO composite aerogel on MB was studied at different temperatures. Table 3 shows the thermodynamic parameters such as the standard enthalpy change (ΔH^Ө^), the average standard entropy change (ΔS^Ө^), and the Gibbs free energy change (ΔG^Ө^), obtained by the Van’t Hoff equation. In the temperature range of 288.15~308.15 K, ΔG^Ө^ is negative, indicating that the adsorption of MB on the aerogel is spontaneous. ΔH^Ө^ is positive, indicating that adsorption is endothermic reaction, high temperature is conducive to the adsorption, which is consistent with the experimental results. ΔS^Ө^ is positive, indicating that the molecular distribution disorder increases with the increase of temperature.

### 4.4. Adsorption Mechanism

The mechanism of adsorption process has been widely researched. Zhu et al. [39] prepared CS/GO aerogel by hydrothermal method, indicating that materials with larger specific surface area and more mesoporous can accommodate more dyes. Yang et al. [40] constructed a new type of 3D graphene aerogel with ultra-high adsorption performance. It is believed that the main adsorption originates from the π-π and electrostatic interactions among adsorbent and target molecules, as well as the synergistic effect of adsorbent polytope surface. Chen et al. [19] synthesized hydrogel to remove dyes, and the adsorption process can be dominated by chemisorption behavior, containing ion exchange and π-π stacking interactions between cationic groups and biomass functional groups.

In this paper, GO aerogel was prepared by EDA reduction, and -NH^3+^ was introduced into graphene sheets. There was a strong electrostatic interaction between polar groups on aerogel and dye molecules, which was beneficial to adsorb anionic dyes. In the preparation process of CMC/GO composite aerogel, carboxymethyl on CMC could form hydrogen bonds and stabilize three-dimensional structure by abundant oxygen-containing groups on graphene oxide sheets [41], and obtain high surface area. In Figure 7, since the ionization of carboxyl groups of CMC, there are a great quantity of negatively charged adsorption point (-COO^−^) on the aerogel surface, resulting in strong electrostatic attraction to cationic dye MB and electron transfer. The molecular size of MB molecule is 1.7 nm ∗ 0.76 nm ∗ 0.33 nm. XRD and BET results show that the interval of layers in CMC/GO composite aerogel is 0.38 nm and the average pore size is 30.13 nm. MB molecules easily enter the pore of the composite aerogel and the middle of the graphene sheet layer, and contact with the adsorption site on the surface of the composite aerogel (with -COOH^-^) to complete the adsorption. As a cationic dye, MB molecule has a positive charge in water. With the increase of doped CMC ratios, the number of -COOH^-^ introduced on the surface of the composite aerogel increases, and the adsorption sites increase accordingly, forming a large amount of electrostatic attraction with the positive charge on the MB molecule. Besides the electrostatic attraction π-π stacking effect, the adsorption mechanism of MB also included hydrogen bonding. During the adsorption process, the amino group in the adsorbent forms hydrogen bonds with MB. The surface structure of GO was grafted with CMC, and the three-dimensional network structure of the composite aerogel remained integrity. Owing to the introduction of CMC, the lamellar spacing and the specific surface area increase, leading to more absorption sites. As a result, CMC/GO composite aerogel shows a better adsorption effect than other reported adsorbents (Table 4), which can well remove MB with high efficiency.

## 5. Conclusions

In this work, the CMC/GO composite aerogel with high large surface area was successfully prepared through hydrothermal method. The structure and properties of this aerogel in relation to the adsorption characteristics were systematically investigated. With the addition of CMC, the agglomerations of graphene sheets are reduced with an increased distance between sheets. At meantime, the specific surface area is increased from 65.01 m^2^·g^−1^ to 800.85 m^2^·g^−1^. The introduced oxygen-containing functional groups induces lots of negatively charged adsorption sites (-COO^-^), leading to an improved hydrophilicity. This composite aerogel shows superior adsorption capacity (244.99 mg·g^−1^) for MB compared to other dyes (RhB, MO, and CR), which presents a strong adsorption selectivity. This aerogel owns a good regeneration performance, because the adsorption rate could still reach up to 92.66% by the rate of first time when subject to nine cycles of adsorption and desorption. Overall, the prepared CMC/GO composite aerogel prepared exhibits an outstanding adsorption capacity for MB and other dyes, which is also proved available to be repeatedly used in high quality. Therefore, it is recommended to use CMC/GO composite aerogel to treat dye pollution in practical application.

## Figures and Tables

**Figure 1 nanomaterials-11-01609-f001:**
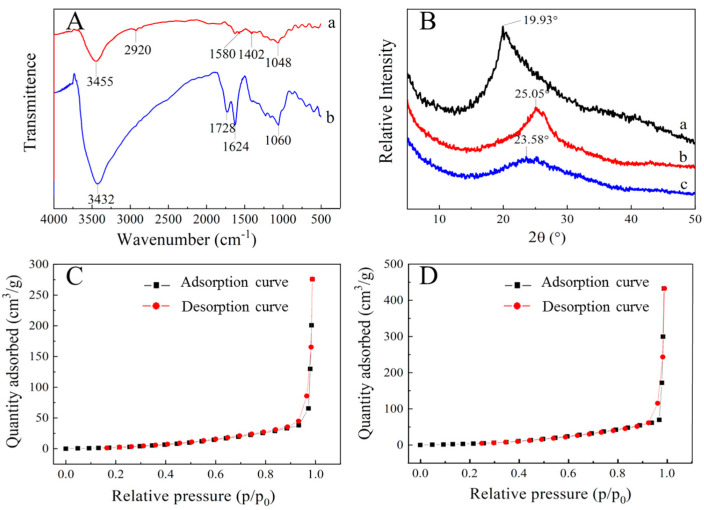
(**A**) FT-IR spectra of (a) CMC/GO aerogel and (b) GO aerogel. (**B**) XRD patterns of aerogels: (a) CMC, (b) GO, and (c) CMC/GO. (**C**) the N_2_ adsorption and desorption curves of GO aerogel. (**D**) the N_2_ adsorption and desorption curves of CMC/GO composite aerogel.

**Figure 2 nanomaterials-11-01609-f002:**
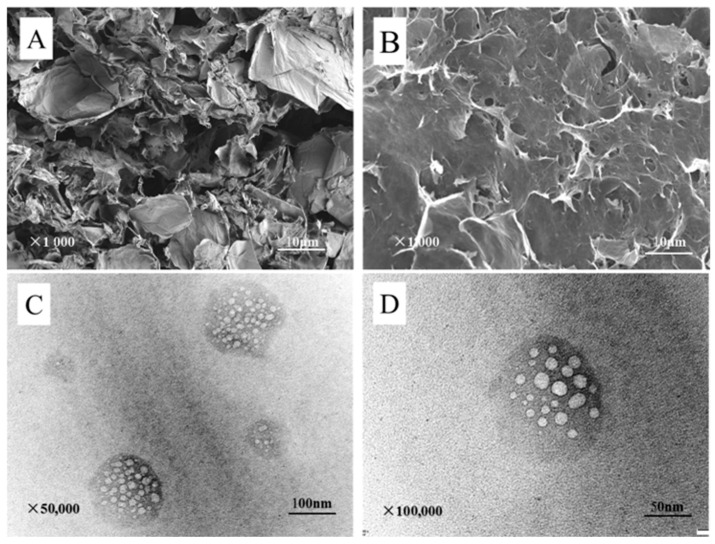
SEM image of aerogels: (**A**) GO and (**B**) CMC/GO. TEM image of CMC/GO (**C**,**D**).

**Figure 3 nanomaterials-11-01609-f003:**
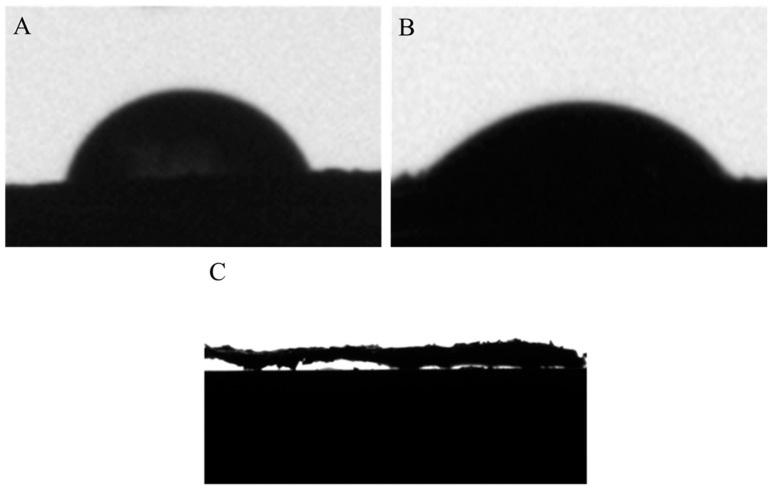
(**A**) advancing contact angle of GO, (**B**) receding contact angle of GO, and (**C**) contact angle of CMC/GO.

**Figure 4 nanomaterials-11-01609-f004:**
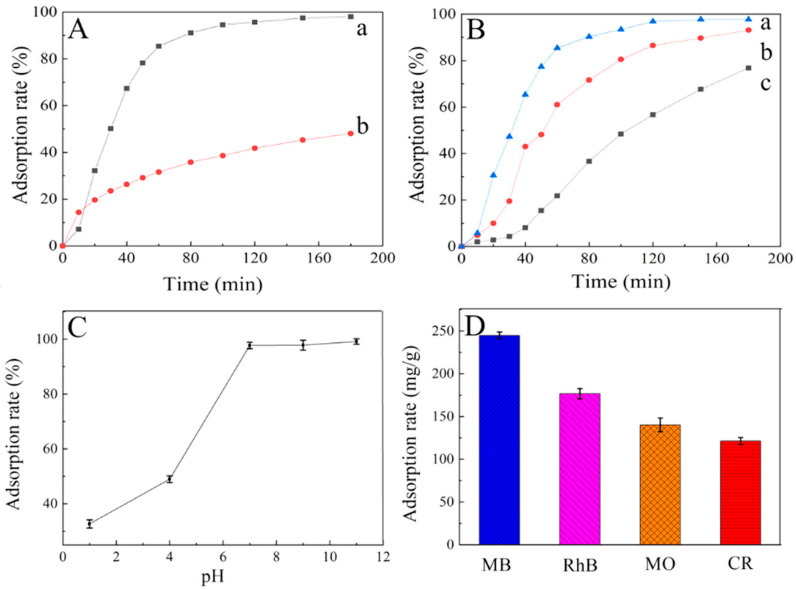
(**A**) adsorption properties of MB on CMC/GO (a) and GO (b) aerogels. (**B**) influence of proportion of materials ratio (m_CMC_:m_GO_) (a = 2:1, b = 1:1, c = 1:2). (**C**) influence of pH. (**D**) adsorption properties of CMC/GO composite aerogel on different dyes.

**Figure 5 nanomaterials-11-01609-f005:**
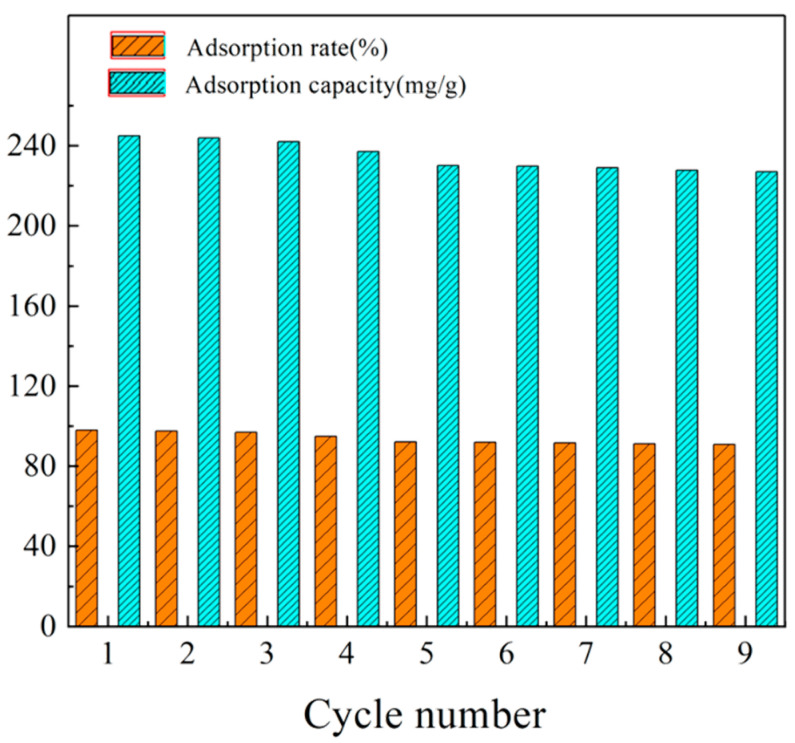
Adsorption rate and adsorption capacity of MB adsorbed by CMC/GO composite aerogel for 9 times.

**Figure 6 nanomaterials-11-01609-f006:**
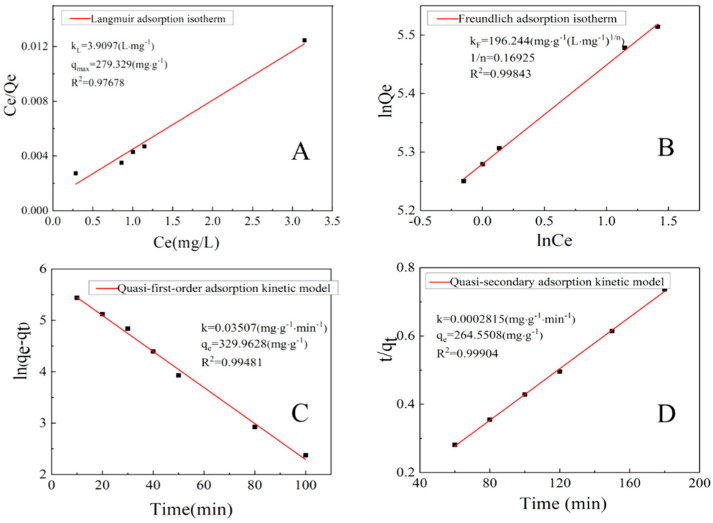
(**A**) Langmuir adsorption isotherm and (**B**) Freundlich adsorption isotherm of CMC/GO. (**C**) fitting diagram of quasi-first-order adsorption kinetics model and (**D**) quasi-secondary adsorption kinetics model.

**Figure 7 nanomaterials-11-01609-f007:**
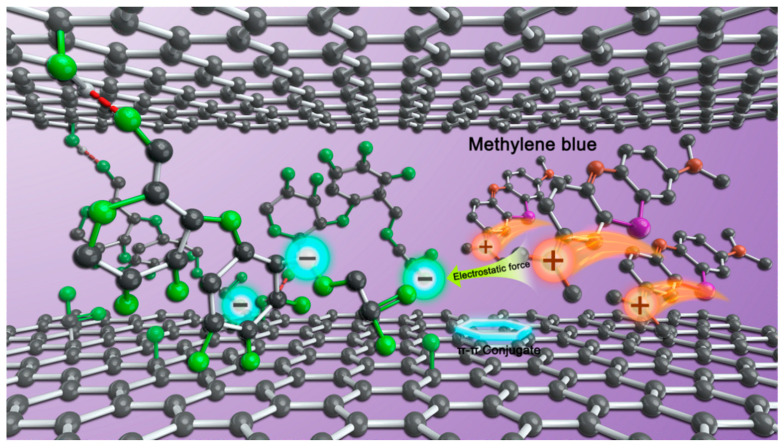
Schematic illustration of the MB adsorption by the CMC/GO composite aerogel.

**Table 1 nanomaterials-11-01609-t001:** Pore structure analysis.

Sample	BET (m^2^·g^−^^1^)	Pore Volume (cm^3^·g^−^^1^)	Pore Diameter (nm)
GO	65.01	0.70	34.09
CMC/GO	800.85	0.45	30.13

**Table 2 nanomaterials-11-01609-t002:** Comparison of circulation performance of different adsorbents.

Adsorbents	Cycle Number	Last Adsorption Capacity/First Adsorption Capacity	References	Date
PVA-based nanocomposite hydrogels	4	93%	[26]	2017
PVA/PCMC/GO/bentonite	4	92.39%	[31]	2018
MnO_2_ nanowires/PU foam composites	5	91.9%	[32]	2016
Starch-humic acid composite hydrogel beads	5	97%	[33]	2015
Magnetic carboxymethyl starch/poly(vinyl alcohol)composite gel	8	85%	[34]	2015
Novel carboxymethyl cellulose/carboxylated graphene oxide composite microbeads	9	90%	[35]	2020
CMC/GO composite aerogel	9	92.66%	This work	2021

**Table 3 nanomaterials-11-01609-t003:** Thermodynamic parameters for the adsorption of MB on CMC/GO composite aerogel.

T(K)	lnK_d_	ΔG^Ө^(KJ·moL^−1^)	ΔH^Ө^(KJ·moL^−1^)	ΔS^Ө^(KJ·moL^−1^·K^−1^)
288.15	1.717	−4.675	10.718	0.0516
298.15	1.83			
308.15	2.01			

**Table 4 nanomaterials-11-01609-t004:** Adsorption performance of different adsorbents for MB.

Adsorbents	Maximum Adsorption Capacity (mg·g^−^^1^)	References	Date
Carboxymethyl cellulose/carboxylated graphene oxide composite microbeads	180.23	[35]	2020
Pineapple peel carboxy methylcellulose-g-poly (acryliccid-co-acrylamide)/graphene oxide hydrogels	133.32	[42]	2019
Polyvinyl alcohol/carboxymethyl cellulose hydrogels	172.14	[31]	2018
Poly(N,N-dimethylacrylamide-co-2-hydroxyethyl methacrylate) hydrogel	80.27	[43]	2018
Pineapple peel cellulose-g-acrylic acid/kaolin/sepia ink hydrogels	153.85	[44]	2017
Modified pineapple peel cellulose hydrogels embedded with sepia ink	138.25	[45]	2016
Starch-humic acid composite hydrogel	110.00	[33]	2015
Guaran/poly(itaconic acid) hydrogel	106.04	[46]	2015
Magnetic starch/poly(vinyl alcohol) composite hydrogel	23.53	[34]	2015
CMC/GO composite aerogel	246.42	This study	2021

## Data Availability

Not applicable.

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
