# Peer review of "Fabrication of Reusable Carboxymethyl Cellulose/Graphene Oxide Composite Aerogel with Large Surface Area for Adsorption of Methylene Blue"

_nanomaterials, 2021, doi:10.3390/nano11061609_

Round 1
Reviewer 1 Report
This paper reports on improvement of adsorption of GO by addition of CMC for MB.
The results were presented clearly, and it is worth publishing in Nanomaterials. There were no serious problems in the text.
However, the reason why this material is suitable for adsorption of MB
was relatively weakly explained as it is. For example, Figure 6 helped
the readers to imagine the circumstance of the adsorption, though
unmentioned forces or factors (e.g. size or shape of MB molecules,
ratios of doped CMC and interval of GO layers and so on) may also
be important for the total role. In this context, the authors should add
detailed explanation of "the reason" closely before acceptance.
That's all.
Reviewer 2 Report
The paper from Zhu et al reports on the synthesis through a one-step hydrothermal method and characterization of carboxymethyl cellulose/graphene oxide composite aerogels with high absorption for methylene blue. The paper is quite well written and the conclusions are quite well supported by the experimental data.
Minor remarks:
- title: "recyclable" should be replaced by "reusable"
- it could be reasonable to perform dynamic contact angle measurements, hence determining the advancing and receding contact angle values and calculating the hysteresis, which can be directly correlated with the surface roughness
- it could be useful to provide some TEM analyses of the synthesized aerogels
